OBSERVATION

# A Novel Widespread MITE Element in the Repeat-Rich Genome of the *Cardinium* Endosymbiont of the Spider *Oedothorax gibbosus*

Tamara Halter,[a,b] Frederik Hendrickx,[c] Matthias Horn,[a] Alejandro Manzano-Marín[a]

aCentre for Microbiology and Environmental Systems Science, University of Vienna, Vienna, Austria
bDoctoral School in Microbiology and Environmental Science, University of Vienna, Vienna, Austria
cOD Taxonomy and Phylogeny, Royal Belgian Institute of Natural Sciences, Brussels, Belgium

**ABSTRACT** Free-living bacteria have evolved multiple times to become host-restricted endosymbionts. The transition from a free-living to a host-restricted lifestyle comes with a number of different genomic changes, including a massive loss of genes. In host-restricted endosymbionts, gene inactivation and genome reduction are facilitated by mobile genetic elements, mainly insertion sequences (ISs). ISs are small autonomous mobile elements, and one of, if not the most, abundant transposable elements in bacteria. Proliferation of ISs is common in some facultative endosymbionts, and is likely driven by the transmission bottlenecks, which increase the level of genetic drift. In this study, we present a manually curated genome annotation for a *Cardinium* endosymbiont of the dwarf spider *Oedothorax gibbosus*. *Cardinium* species are host-restricted endosymbionts that, similarly to *ColbachiaWolbachia* spp., include strains capable of manipulating host reproduction. Through the focus on mobile elements, the annotation revealed a rampant spread of ISs, extending earlier observations in other *Cardinium* genomes. We found that a large proportion of IS elements are pseudogenized, with many displaying evidence of recent inactivation. Most notably, we describe the lineage-specific emergence and spread of a novel IS-derived Miniature Inverted repeat Transposable Element (MITE), likely being actively maintained by intact copies of its parental IS982-family element. This study highlights the relevance of manual curation of these repeat-rich endosymbiont genomes for the discovery of novel MITEs, as well as the possible role these understudied elements might play in genome streamlining.

**IMPORTANCE** *Cardinium* bacteria, a widespread symbiont lineage found across insects and nematodes, have been linked to reproductive manipulation of their hosts. However, the study of *Cardinium* has been hampered by the lack of comprehensive genomic resources. The high content of mobile genetic elements, namely, insertion sequences (ISs), has long complicated the analyses and proper annotations of these genomes. In this study, we present a manually curated annotation of the *Cardinium* symbiont of the spider *Oedothorax gibbosus*. Most notably, we describe a novel IS-like element found exclusively in this strain. We show that this mobile element likely evolved from a defective copy of its parental IS and then spread throughout the genome, contributing to the pseudogenization of several other mobile elements. We propose this element is likely being maintained by the intact copies of its parental IS element and that other similar elements in the genome could potentially follow this route.

**KEYWORDS** *Cardinium*, mobile element, insertion sequence, endosymbiont, *Amoebophilaceae*

Address correspondence to Alejandro Manzano-Marín, alejandro.manzano.marin@univie.ac.at.

The authors declare no conflict of interest.

Microbial symbionts are widespread across the animal kingdom, shaping their hosts' evolution and serving them as a source for novel metabolic capabilities (1, 2). Of particular interest are those relations that evolve between microbes and their hosts where the

microbe cannot thrive outside of its host's cells or tissues. Within these endosymbionts, we find the widely studied obligate nutritional symbioses observed in phloem- or blood-feeders as well as those involving facultative endosymbiotic lineages (3). A widespread feature of the genomes of facultative endosymbionts is the presence of large numbers of mobile elements; including prophages, group II introns, and mainly insertion sequences (ISs) (4–8). In their simplest form, ISs are small, autonomous, transposable elements encoding for a transposase gene flanked by terminal inverted repeats. ISs are arguably the most numerous mobile elements in bacteria (9), and have been implicated in promoting widespread genome rearrangement and differential pseudogenization between closely related strains of facultative endosymbionts (10–14).

Facultative endosymbionts include both conditional beneficial symbionts as well as reproductive manipulators. Reproductive manipulation entails, most famously, male killing, feminization, and cytoplasmic incompatibility (*CI*) (15), which can facilitate the spread of the endosymbiont in a host population (16). While *Wolbachia* strains are the most notorious male killing and *CI*-inducing endosymbionts, specific strains of the endosymbiotic genus *Cardinium* have also been shown to be involved in inducing *CI* (17), parthenogenesis (18), and feminization (19). Analysis of the genome of a *CI*-inducing *Cardinium* strain from the parasitoid wasp *Encarsia pergandiella* suggests an independent evolution of the CI phenotype in *Wolbachia* and *Cardinium* lineages (20). Based on available genome sequences, *Cardinium* strains have been organized into 3 groups, with group A containing exclusively endosymbionts from arthropods (namely, insects and mites), group B from nematodes, and group C from *Culicoides punctatus* (Diptera: Ceratopogonidae) (21). Similarly to *Wolbachia*, phylogenetic analyses suggest that occasional switching between distant host phyla may be a feature of the genus *Cardinium* (21). To date, 8 genomes of different finishing status are available in the databases. All strains hold genomes of around 1 Mega base-pair (Mbp) and are highly enriched in mobile elements, namely, ISs. Despite the important role IS elements play in both genome inactivation and genome rearrangement, only the cBtQ1 strain isolated from the whitefly *Bemisia tabaci* biotype MEDQ1 has undergone rigorous annotation of these elements (22). In this study, we present the manually curated annotation of the *Cardinium* endosymbiont of the spider *Oedothorax gibbosus*. This revealed an abundant small non-autonomous IS-derived Miniature Inverted repeat Transposable Element (or MITE), which to our knowledge, is previously unreported for endosymbionts and is unique to this *Cardinium* strain.

The genome of *Cardinium* strain cOegibbosus-W744x776 (hereafter cOegib) was assembled previously from long- and short-read data generated for the genome sequencing of its spider host, *O. gibbosus* (23, 24). To produce a high-quality annotation of the mobile elements of cOegib, an initial draft annotation was done using Prokka v1.14.6 (25). This draft annotation was followed by careful manual curation using a combination of DELTA-BLASTP (versus NCBI's nr and Swiss-Prot) (26), InterProScan v5.45-80.0 (27), Infernal v1.1.3 (–cut_tc –mid; versus Rfam v14.2) (28, 29), tRNAscan-SE v2.0.9 (-B –isospecific) (30), and ARAGORN v1.2.38 (31). Finally, careful manual searches against the ISfinder database (32) were performed in order to identify complete, partial, and fragmented elements across the genome, with special care to correctly identify the terminal inverted repeats of IS elements.

As previously reported in Halter et al. (24), the general features of the genome of cOegib are comparable to other members of the *Cardinium* genus (Table S1). Similarly to other *Cardinium* strains (20, 22), a large fraction of cOegib's genome (29.60%) is made up of mobile elements, which is the highest reported for the genus. Manual curation revealed that group II introns and ISs made up the majority of these elements (87.97%), with the latter being by far the most abundant. From the repertoire of in total 300 ISs, we were able to identify ISCca2 to ISCca6 elements, previously reported for other *Cardinium* strains (20, 22). Nonetheless, their copy numbers are dissimilar to those of strain cBtQ1, revealing lineage-specific expansions/contractions (Table 1). In addition to these *Cardinium*-specific ISs, we were able to detect an additional 8 mobile elements (named ISCca9 to ISCca16) belonging to 7 different IS families. Despite being highly abundant, only 23.33% of these IS elements, belonging to 6 different types, encode for an intact transposase gene, and 8.00% are only partial IS elements with no transposase gene/pseudogene. This suggests that only the subset of

**TABLE 1** Distribution of intact and partial IS elements in selected *Cardinium*

| Is type | Family, group | cOegib | | cBtQ1 | |
|---|---|---|---|---|---|
| | | Total | Intact | Total | Intact |
| ISCca1[a] | IS982 | 0 | 0 | 28 | 8 |
| ISCca2 | IS6 | 109 | 43 | 18 | 4 |
| ISCca3 | IS5, IS5 | 20 | 10 | 5 | 2 |
| ISCca4 | IS982 | 68 | 7 | 41 | 10 |
| ISCca5 | IS5, IS5 | 3 | 0 | 38 | 2 |
| ISCca6 | IS1634 | 4 | 0 | 5 | 2 |
| ISCca7[+] | IS5, IS5 | 0 | 0 | 10 | 1 |
| ISCca8[−] | IS6 | 0 | 0 | 1 | 0 |
| ISCca9 | IS256 | 35 | 2 | 0 | 0 |
| ISCca10 | IS256 | 26 | 4 | 0 | 0 |
| ISCca11 | IS3, IS150 | 11 | 0 | 0 | 0 |
| ISCca12 | IS110 | 9 | 0 | 0 | 0 |
| ISCca13 | IS4, IS4 | 9 | 4 | 0 | 0 |
| ISCca14 | IS481 | 4 | 0 | 0 | 0 |
| ISCca15 | Tn3 | 1 | 0 | 0 | 0 |
| ISCca16 | IS66 | 1 | 0 | 0 | 0 |

[a]ISCca1 is highly similar to ISCca4. [+], ISCca7 is highly similar to ISCca3. [−], ISCca8 is highly similar to ISCca2. These two *Cardinium* strains were selected given that they are the only two to undergo thorough manual curation for IS elements.

IS elements that have at least one intact copy in the genome still preserve the ability to mobilize, while those that do not are likely to be eventually purged, unless new intact copies are acquired through horizontal gene transfer. This hypothesis is supported by the 5 most abundant IS elements matching all but one of the intact ones.

Most notably, manual curation of the IS elements revealed a large number of a shorter sequence of *circa* 240 bp flanked by an inverted repeat very similar to those of ISCca4 (IS982 family) (Fig. 1A), the second most abundant IS element in the cOegib genome. However, these shorter repeats completely lack a transposase gene. Upon closer inspection, we found good evidence to suggest that these shorter repeats are likely derived from a parental ISCca4 element: downstream of the left inverted repeat, there is a short 5 bp-long conserved sequence when compared to ISCca4. In addition, despite MITECca01 possessing identical but shorter inverted repeats, it preserves the 5′-AGMTTGTW-3′ downstream sequence from its likely parental ISCca4. This novel *Cardinium* IS-like element, designated MITECca01, has all the features of Miniature Inverted repeat Transposable Elements (or MITEs), which are short (typically shorter than 300 bp), non-autonomous transposable elements that depend on a functional transposase gene of their parental IS to mobilize (9). Hitherto, MITEs have been reported in animals, protists, fungi, and a few plants, bacteria, and archaea (9, 33), with no reports, to our knowledge, in maternally-inherited intracellular endosymbionts. This underreporting might be due to their small nature, a lack of thorough annotation of mobile genetic elements in genomes, and the lack of available full-length annotations (i.e., including terminal inverted repeats) of their parental IS elements.

The newly identified MITE has successfully spread throughout the genome of cOegib, effectively becoming the most abundant mobile element in the genome, with 169 copies and making up 3.5% of the total chromosomal sequence. This large number of copies contrasts even the 5 most abundant IS elements, which are present in 109 (ISCca2), 68 (ISCca4), 35 (ISCca9), 26 (ISCca10), and 20 (ISCca3) copies, respectively. Similar to other ISs in the genome of cOegib and other endosymbionts (11, 13, 14), MITECca01 was found mostly in intergenic regions as well as disrupting other mobile elements, and much less commonly inactivating protein-coding genes. Its genomic distribution points toward a role for MITECca01 in IS element inactivation in the genome of cOegib. Contrary to IS elements, very few MITECca01 elements are found disrupted by other mobile elements, which could likely be due to their small size compared to ISs (240 versus ca. 1000 bp). Further, we were able to identify a second MITE element very similar to MITECca01, termed MITECca02 (Fig. 1B), which likely represents yet another MITE element with potential to spread throughout

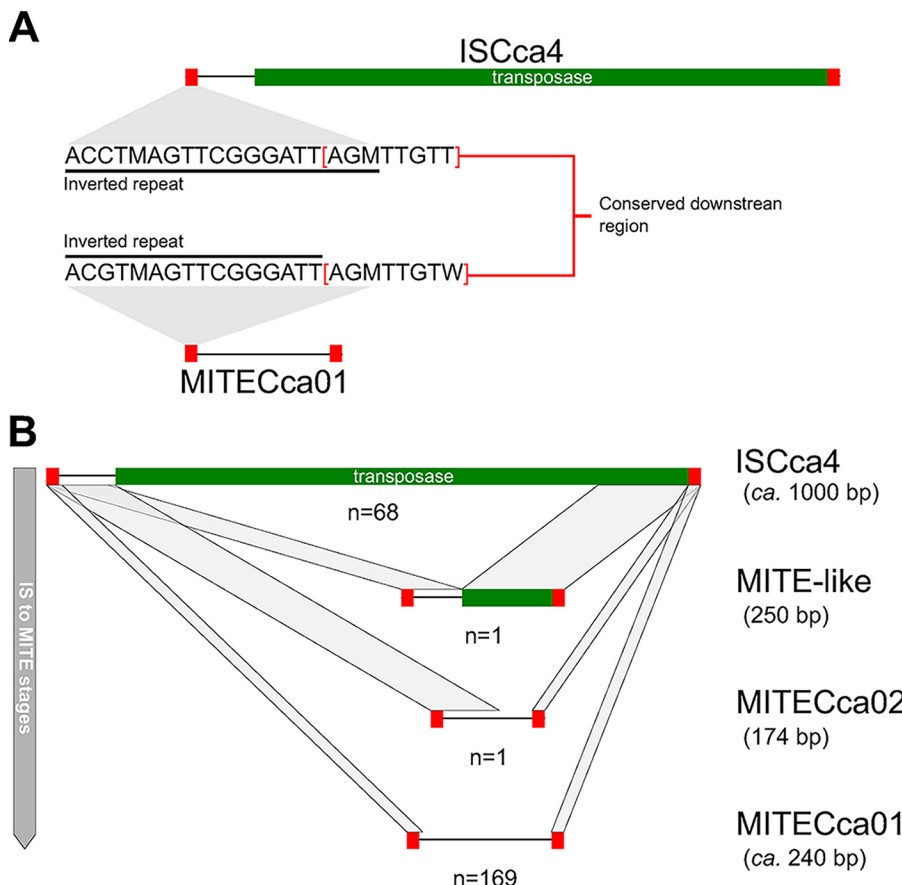

**FIG 1** Diagram depicting ISCca4, MITE-like, and MITE elements detected in cOegib. In red boxes, inverted repeats of each element are displayed in both subpanels. (A) Comparison of ISCca4 and MITECca01 and their inverted repeats. Sequence of the palindromic inverted repeats are shown underlined in black. Red square brackets highlight the conserved sequence downstream of the left inverted repeats of both mobile elements. (B) Diagram depicting, form top to bottom, the ISCca4 and related MITEs in order of likely stages of MITE formation and copy number increase across the genome. At the bottom-right of each panel, the number of copies is expressed as "n" under the diagram of each element. Gray parallelograms connecting the linear depiction of the mobile elements indicate conserved regions.

this *Cardinium* lineage. This second MITE element might be evolutionarily younger, given its copy number and the fact it keeps a much longer conserved sequenced with ISCca4 following the left inverted repeat (Fig. S1). On top of these 2 MITE elements, we also identified an uninterrupted highly eroded ISCca4 element of 250 bp in length that retains a small part of the 3′-end of its transposase gene (Fig. S1). This IS remnant potentially represents a very early stage of the birth of a MITE element. Finally, blastn searches of these novel *Cardinium* MITE elements did not reveal the presence of these in any of the other sequenced *Cardinium* strains, suggesting that it originated in the lineage leading to cOegib.

In conclusion, through the thorough annotation of the *Cardinium* strain cOegib, we were able to shed light on the dynamics that likely resulted in the current distribution and abundance of mobile elements in this genome. In addition, the MITECca01 element present in the cOegib genome represents, to our knowledge, a novel kind of IS-like element in a maternally-inherited endosymbiont lineage. This novel MITE has been very successful in multiplying across the genome, and its mobility and persistence is likely linked to the survival of intact copies of the "parental" ISCca4 element. Finally, the genome of cOegib along with its careful and detailed annotation represents a valuable resource with relevance to the continued study of this endosymbiont taxon and transposable elements, as well as the larger field of reductive genome evolution.

**Data availability.** The genome annotation of the *Cardinium* endosymbiont of *Oedothorax gibbosus* strain cOegibbosus-W744x776 has been deposited in the European Nucleotide Archive (ENA) under accession number OW441264.

## SUPPLEMENTAL MATERIAL

Supplemental material is available online only.
**SUPPLEMENTAL FILE 1**, XLS file, 0.01 MB.
**SUPPLEMENTAL FILE 2**, PDF file, 0.1 MB.

## ACKNOWLEDGMENTS

This project has received funding from the University of Vienna (uni:docs to T.H.) and the Austrian Science Fund FWF (DOC 69-B). A.M.-M. was supported by the European Union's Horizon 2020 research and innovation program under a Marie Skłodowska-Curie Individual Fellowship (LEECHSYMBIO, grant agreement no. 840270). F.H. was supported by an Individual Research Grant (Fund for Scientific Research – 152761N).

The funders had no role in study design, data collection and analysis, decision to publish, or preparation of the manuscript.

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
