## [Reviewer comments · Microbiology Spectrum]

Microbiology Spectrum

A novel widespread MITE element in the repeat-rich genome of the *Cardinium* endosymbiont of the spider *Oedothorax gibbosus*

Tamara Halter, Frederik Hendrickx, Matthias Horn, and Alejandro Manzano-Marín

Corresponding Author(s): Alejandro Manzano-Marín, University of Vienna

Review Timeline:

Submission Date:	July 10, 2022
Editorial Decision:	August 17, 2022
Revision Received:	September 7, 2022
Accepted:	September 19, 2022

Editor: Swaine Chen

Reviewer(s): The reviewers have opted to remain anonymous.

Transaction Report:

DOI: <https://doi.org/10.1128/spectrum.02627-22>

August 17, 2022

Dr. Alejandro Manzano-Marín
University of Vienna
Centre for Microbiology and Environmental Systems Science
Djerassiplatz 1
Vienna 1030
Austria

Re: Spectrum02627-22 (A novel widespread MITE element in the repeat-rich genome of the *Cardinium* endosymbiont of the spider *Oedothorax gibbosus*)

Dear Dr. Alejandro Manzano-Marín:

Link Not Available

Sincerely,

Journals Department
Reviewer comments:

Reviewer #1 (Comments for the Author):

The manuscript "A novel widespread MITE element in the repeat-rich genome of the *Cardinium* endosymbiont of the spider *Oedothorax gibbosus*" by Halter et al. presents a detailed annotation of mobile genetic elements (MGEs) in the genome of a host-restricted *Cardinium* endosymbiont. Thorough annotations of MGEs are uncommon in host-restricted endosymbionts, despite the potentially important role that they play in genome evolution and streamlining in these organisms, making this an important contribution to the field. The methods used to annotate MGEs are sound, and I think this work would be well suited for publication in Microbiology Spectrum.

The main findings are that insertion sequences (ISs), one of the most abundant types of MGEs in bacterial genomes, are abundant and frequently pseudogenised in the studied genome assembly. Specific to the *Cardinium* lineage, the authors describe a novel IS-derived Miniature Inverted repeat Transposable Element (MITE), MITECca01. By examining sequence similarity, the authors demonstrate that MITECca01 is the MGE with the highest copy number in the genome and is likely a truncated version of a longer MGE (ISCca4). Unlike MITEs, ISCca4 is predicted to encode a functional transposase and retain the capacity to transpose autonomously. The authors suggest that the transposition activity of ISCca4 maintains copies of MITECca01 in the genome and enables their proliferation. MITECca01 was distributed largely in intergenic regions and other MGEs, and less so in protein coding genes, which the authors hypothesise suggests a role of MITECca01 in inactivating other MGEs and driving genome streamlining.

The manuscript is well written and clear. I don't have any major comments, but a few minor comments that I think would improve the manuscript are as follows:

Lines 9-11: In my opinion, for the authors to propose that autonomous copies of ISCca4 maintain MITECca01 elements in the genome, quite a bit of laboratory/in vivo work would be required. I am not at all suggesting that would be appropriate for this manuscript, but it would be nice to see this addressed in the discussion how the authors would propose to do this. e.g. are the ISCca4 transposases expressed? Do you detect transposase protein (tough with low abundance proteins, but maybe possible), are they capable of transposing MITECca01?

Lines 80-82: The short 5 bp-long conserved sequence- I feel that this could be explained a little better. Does this mean this sequence is conserved across all MITECca01 and all ISCca4 copies? Could it be shown on Fig. 1? Currently I don't see it there, but maybe I am misreading the figure. As this seems to be the main piece of evidence for the claim in the 'Importance' section on lines 7-9 that "We show that this mobile element likely evolved from a defective copy of its parental IS and then spread throughout the genome, contributing to pseudogenisation of several other mobile genetic elements", I think this warrants further clarification.

Lines 86-87: I believe MITEs are also common in animals, protists and fungi.

Line 110: Suggest changing "shade light" to "shed light".

Reviewer #2 (Comments for the Author):

The study describes mobile genetic elements, e.g. insertion sequences. The study is focused on the lineage-specific emergence and spread of a novel IS-derived Miniature Inverted repeat Transposable Element (MITE) in intracellular symbiont *Cardinium* in the spider *Oedothorax gibbosus*. The Ms has a lot of novelty aspects and belongs to the aim and scope of the journal. The authors claim that MITE was exclusively founded in *Cardinium* strain. The MS is well written and I have read it more times, but I have just few recommendation of minor importance.

For lines 50 -60 I suggest putting the material and methods section from this chapter.

References - I suggest adding the references for compared genomes in Table S1 into the references of this draft. I'm missing the author contribution statements in the draft.

Staff Comments:

Preparing Revision Guidelines

Please return the manuscript within 60 days; if you cannot complete the modification within this time period, please contact me. If

you do not wish to modify the manuscript and prefer to submit it to another journal, please notify me of your decision immediately so that the manuscript may be formally withdrawn from consideration by Microbiology Spectrum.

Response to Reviewers

Reviewer #1

The manuscript "A novel widespread MITE element in the repeat-rich genome of the *Cardinium* endosymbiont of the spider *Oedothorax gibbosus*" by Halter *et al.* presents a detailed annotation of mobile genetic elements (MGEs) in the genome of a host-restricted *Cardinium* endosymbiont. Thorough annotations of MGEs are uncommon in host-restricted endosymbionts, despite the potentially important role that they play in genome evolution and streamlining in these organisms, making this an important contribution to the field. The methods used to annotate MGEs are sound, and I think this work would be well suited for publication in *Microbiology Spectrum*.

-> Thank you for the positive comment.

The main findings are that insertion sequences (ISs), one of the most abundant types of MGEs in bacterial genomes, are abundant and frequently pseudogenised in the studied genome assembly. Specific to the *Cardinium* lineage, the authors describe a novel IS-derived Miniature Inverted repeat Transposable Element (MITE), MITECca01. By examining sequence similarity, the authors demonstrate that MITECca01 is the MGE with the highest copy number in the genome and is likely a truncated version of a longer MGE (ISCca4). Unlike MITEs, ISCca4 is predicted to encode a functional transposase and retain the capacity to transpose autonomously. The authors suggest that the transposition activity of ISCca4 maintains copies of MITECca01 in the genome and enables their proliferation. MITECca01 was distributed largely in intergenic regions and other MGEs, and less so in protein coding genes, which the authors hypothesise suggests a role of MITECca01 in inactivating other MGEs and driving genome streamlining.

The manuscript is well written and clear. I don't have any major comments, but a few minor comments that I think would improve the manuscript are as follows:

Lines 9-11: In my opinion, for the authors to propose that autonomous copies of ISCca4 maintain MITECca01 elements in the genome, quite a bit of laboratory/in vivo work would be required. I am not at all suggesting that would be appropriate for this manuscript, but it would be nice to see this addressed in the discussion how the authors would propose to do this. e.g. are the ISCca4 transposases expressed? Do you detect transposase protein (tough with low abundance proteins, but maybe possible), are they capable of transposing MITECca01?

-> We agree with the reviewer that in order to actually prove that transposases encoded in ISCca4 elements mobilise MITECca01 laboratory work is needed, likely in an experimental evolution setting. We believe that proving the expression of the transposase gene of ISCca4 would do no more beyond proving that the copies of this mobile element with an intact transposase gene are actually expressing it. This expression would not be surprising, given the predicted annotation of an intact IS element. In order to prove the hypothesis laid down here, a setting where one could block or knock out the transposase genes of all seven intact ISCca4 elements in cOegib. Then, one would evolve the "wild-type" and the mutant for X number of generations and

would then compare the genomes. If the hypothesis holds, one would expect to see the “wild-type” line displaying some to many more MITE mobilisations than the mutant. The problem with this setting is that the (likely) inability to stably culture this *Cardinium* strain, genetically modify it, and the unknown time/generations for new insertions to happen would make this experiment likely to fail. We believe the high copy-number of MITECca01 and the conservation of the inverted repeats with ISCca4, provides sufficient evidence for the emergence of a single MITECca01 followed by mobilisation of it by an ISCca4 transposase. We have nonetheless been careful to only word this as the most likely hypothesis across the manuscript, following the reviewer’s comment.

Lines 80-82: The short 5 bp-long conserved sequence- I feel that this could be explained a little better. Does this mean this sequence is conserved across all MITECca01 and all ISCca4 copies? Could it be shown on Fig. 1? Currently I don't see it there, but maybe I am misreading the figure. As this seems to be the main piece of evidence for the claim in the 'Importance' section on lines 7-9 that "We show that this mobile element likely evolved from a defective copy of its parental IS and then spread throughout the genome, contributing to pseudogenisation of several other mobile genetic elements", I think this warrants further clarification.

-> The 5-bp sequence is indeed conserved across MITECca01 ISCca4. While ISCca4's inverted repeats have the sequence 5'-ACCTMAGTTCGGGATTAGM-3' (underlined in figure 1 and labeled “Inverted repeat”) followed by the 5-bp sequence 5'-TTGTT-3'. The MITECca01 (as well as MITECca02) keep a shorter version of the inverted repeat 5'-ACCTMAGTTCGGGATT-3' (also underlined in figure 1 and labeled “Inverted repeat”) from ISCca4 lacking the “AGM” 3'-terminal sequence from ISCca4. The evidence from the MITECca01 element originating from ISCca4 lies within this aforementioned “AGM” sequence plus the 5'-TTGTW-3' (W being either A or T). Therefore, and despite MITECca01 having slightly shorter inverted repeats identical to ISCca4, it keeps the “AGMTTGTW” sequence immediately downstream of the left inverted repeat. This region is highlighted in red brackets and labeled “Conserved downstream region” in Figure 1A. Following the reviewer’s comment, we have made the explanation clearer in the main text and added also explanations in Figure 1 for the red boxes and shaded parallelograms connecting the diagrams of mobile elements.

Lines 86-87: I believe MITES are also common in animals, protists and fungi.

-> They indeed are. This has now been stated in the manuscript.

Line 110: Suggest changing "shade light" to "shed light".

-> Changed... shade was definitely not the right word here.

Reviewer #2

The study describes mobile genetic elements, e.g. insertion sequences. The study is focused on the lineage-specific emergence and spread of a novel IS-derived Miniature Inverted repeat Transposable Element (MITE) in intracellular symbiont *Cardinium* in the spider *Oedothorax gibbosus*. The Ms has a lot of novelty aspects and belongs to the aim and scope of the journal. The authors claim that MITE was exclusively founded in

Cardinium strain. The MS is well written and I have read it more times, but I have just few recommendations of minor importance.

-> We thank the reviewer for his/her kind comment.

For lines 50 -60 I suggest putting the material and methods section from this chapter.

-> Since this is an "Observation" type paper, the format only includes one main section. In this case, we have opted to weave the materials and methods in the main Results and Discussion section every time before going into the results.

References - I suggest adding the references for compared genomes in Table S1 into the references of this draft.

-> They have been added.

I'm missing the author contribution statements in the draft.

-> The author contributions are filled up when submitting the paper. They are filled according to a set of roles on the online system. There is no place in the format to include them in the manuscript. Therefore, they cannot be added onto the manuscript text.

September 19, 2022

Dr. Alejandro Manzano-Marín
University of Vienna
Centre for Microbiology and Environmental Systems Science
Djerassiplatz 1
Vienna 1030
Austria

Re: Spectrum02627-22R1 (A novel widespread MITE element in the repeat-rich genome of the *Cardinium* endosymbiont of the spider *Oedothorax gibbosus*)

Dear Dr. Alejandro Manzano-Marín:

Your manuscript has been accepted, and I am forwarding it to the ASM Journals Department for publication. You will be notified when your proofs are ready to be viewed.

Sincerely,

Journals Department
Supplemental table S1: Accept
Supplemental figure S1: Accept